# Sequencing the Genomes of the First Terrestrial Fungal Lineages: What Have We Learned?

**DOI:** 10.3390/microorganisms11071830

**Published:** 2023-07-18

**Authors:** Andrii P. Gryganskyi, Jacob Golan, Anna Muszewska, Alexander Idnurm, Somayeh Dolatabadi, Stephen J. Mondo, Vira B. Kutovenko, Volodymyr O. Kutovenko, Michael T. Gajdeczka, Iryna M. Anishchenko, Julia Pawlowska, Ngoc Vinh Tran, Ingo Ebersberger, Kerstin Voigt, Yan Wang, Ying Chang, Teresa E. Pawlowska, Joseph Heitman, Rytas Vilgalys, Gregory Bonito, Gerald L. Benny, Matthew E. Smith, Nicole Reynolds, Timothy Y. James, Igor V. Grigoriev, Joseph W. Spatafora, Jason E. Stajich

**Affiliations:** 1Division of Biological & Nanoscale Technologies, UES, Inc., Dayton, OH 45432, USA; 2Department of Botany, University of Wisconsin-Madison, Madison, WI 53706, USA; jjg384@nyu.edu; 3Institute of Biochemistry & Biophysics, Polish Academy of Sciences, 01-224 Warsaw, Poland; ania.muszewska@gmail.com; 4School of BioSciences, University of Melbourne, Parkville, VIC 3010, Australia; alexander.idnurm@unimelb.edu.au; 5Biology Department, Hakim Sabzevari University, Sabzevar 96179-76487, Iran; somayeh99@gmail.com; 6U.S. Department of Energy Joint Genome Institute, Lawrence Berkeley National Laboratory, Berkeley, CA 94720, USA; sjmondo@lbl.gov (S.J.M.); ivgrigoriev@lbl.gov (I.V.G.); 7Department of Agrobiology, National University of Life & Environmental Sciences, 03041 Kyiv, Ukraine; virakutovenko@gmail.com (V.B.K.);; 8LF Lambert Spawn Co., Coatesville, PA 19320, USA; mgajdeczka@gmail.com; 9MG Kholodny Institute of Botany, National Academy of Sciences, 01030 Kyiv, Ukraine; ira_anishchenko@hotmail.com; 10Institute of Evolutionary Biology, Faculty of Biology, Biological & Chemical Research Centre, University of Warsaw, 02-089 Warsaw, Poland; julia.z.pawlowska@uw.edu.pl; 11Plant Pathology Department, University of Florida, Gainesville, FL 32611, USA; tran@bio.uni-frankfurt.de (N.V.T.); gbenny@ufl.edu (G.L.B.); trufflesmith@gmail.com (M.E.S.); 12Leibniz Institute for Natural Product Research & Infection Biology, 07745 Jena, Germany; ebersberger@bio.uni-frankfurt.de (I.E.); kerstin.voigt@leibniz-hki.de (K.V.); 13Department of Ecology & Evolutionary Biology, University of Toronto, Toronto, ON M5S 1A1, Canada; yanxw.wang@utoronto.ca; 14Department of Biological Sciences, University of Toronto Scarborough, Toronto, ON M1C 1A4, Canada; 15Department of Biological Sciences, National University of Singapore, Singapore 119077, Singapore; ying.chang@yale-nus.edu.sg; 16School of Integrative Plant Science, Cornell University, Ithaca, NY 14850, USA; tep8@cornell.edu (T.E.P.); nr299@cornell.edu (N.R.); 17Department of Molecular Genetics & Microbiology, Duke University School of Medicine, Durham, NC 27710, USA; heitm001@duke.edu; 18Biology Department, Duke University, Durham, NC 27708, USA; fungi@duke.edu; 19Department of Plant, Soil & Microbial Sciences, Michigan State University, East Lansing, MI 48824, USA; bonito@msu.edu; 20Department of Ecology & Evolutionary Biology, University of Michigan, Ann Arbor, MI 48109, USA; tyjames@umich.edu; 21Department of Plant & Microbial Biology, University of California, Berkeley, CA 94720, USA; 22Department of Botany & Plant Pathology, Oregon State University, Corvallis, OR 97331, USA; spatafoj@science.oregonstate.edu; 23Department of Microbiology & Plant Pathology, University of California, Riverside, CA 93106, USA; jason.stajich@ucr.edu

**Keywords:** fungal evolution, ecological relevance, saprotrophs, pathogens, model organisms

## Abstract

The first genome sequenced of a eukaryotic organism was for *Saccharomyces cerevisiae*, as reported in 1996, but it was more than 10 years before any of the zygomycete fungi, which are the early-diverging terrestrial fungi currently placed in the phyla *Mucoromycota* and *Zoopagomycota*, were sequenced. The genome for *Rhizopus delemar* was completed in 2008; currently, more than 1000 zygomycete genomes have been sequenced. Genomic data from these early-diverging terrestrial fungi revealed deep phylogenetic separation of the two major clades—primarily plant—associated saprotrophic and mycorrhizal *Mucoromycota* versus the primarily mycoparasitic or animal-associated parasites and commensals in the *Zoopagomycota*. Genomic studies provide many valuable insights into how these fungi evolved in response to the challenges of living on land, including adaptations to sensing light and gravity, development of hyphal growth, and co-existence with the first terrestrial plants. Genome sequence data have facilitated studies of genome architecture, including a history of genome duplications and horizontal gene transfer events, distribution and organization of mating type loci, rDNA genes and transposable elements, methylation processes, and genes useful for various industrial applications. Pathogenicity genes and specialized secondary metabolites have also been detected in soil saprobes and pathogenic fungi. Novel endosymbiotic bacteria and viruses have been discovered during several zygomycete genome projects. Overall, genomic information has helped to resolve a plethora of research questions, from the placement of zygomycetes on the evolutionary tree of life and in natural ecosystems, to the applied biotechnological and medical questions.

## 1. Introduction

The first “land” fungi retained their flagella for motility and remained adapted to life in water, and several diverging lineages switched to multicellularity while exploiting new environments [1]. A hallmark of the process of adapting to terrestrial life was the loss of the flagellum, which occurred independently several times during fungal evolution, giving rise to fungi with a hyphal growth form [2]. Prominent among early-divergent fungi is a group of lineages historically known as the zygomycetes, which are unified by similarities in reproductive biology (e.g., the formation of sexual zygospores via undifferentiated gametangia) and mostly do not form macroscopic fruiting bodies, making them more difficult to study than *Dikarya* [3]. Zygomycetes have also been challenging to study because many species are involved in obligate symbiotic interactions and have not been successfully grown in pure culture [4]. However, subsequent molecular studies have shown that the zygomycete fungi are paraphyletic [5]. Today, zygomycetes are recognized as two independent monophyletic groups divided into six subphyla [6]. Terrestrial environments posed novel challenges to early fungi, and the adaptive solutions that these lineages evolved can today be gleaned from the vast library of genes contained within their genomes [7,8]. While these first land colonizers are now extinct, many of their features are preserved in the genomes of extant zygomycetes. One instance, in which whole-genome data have proven useful, is in understanding the origin and evolution of fungi. About 450 million years ago, combined activities of fungi and plants placed the terrestrial world on a trajectory toward its current state, setting the stage for subsequent evolutionary transitions. Primitive mycorrhiza-like relationships and other forms of fungal–plant associations helped facilitate the successful colonization of land by the first terrestrial plants, and the genes responsible for such interactions are still preserved in fungal genomes [9]. Hence, zygomycete genomes have and will continue to play an integral role in our current understanding of the initial transition of organisms to land [10].

The number of sequenced fungal genomes continues to grow exponentially, and genomic data have become widely and freely accessible. While most sequenced genomes are from *Ascomycota* and *Basidiomycota* (*Dikarya*) species, there have been concerted efforts to sequence genomes of early-diverging and underrepresented lineages [11,12,13,14,15]. Since 2014, the number of zygomycete genomes has grown from a dozen [16] to nearly a thousand. However, 70% of these genomes are still in the draft stage, and not all of them are available. While the number of species with sequenced genomes for all organisms is on the rise, the available pool of zygomycete genomes is exceptional due to its diversity, i.e., one-third of ca. 1000 known zygomycete species have already been sequenced (Appendix A [17]).

The sequencing efforts have not been uniform among different groups of *Mucoromycota* and *Zoopagomycota*; for most species, only a single representative genome is available. While several major clades lack a single published genome (e.g., *Eryniopsis* and *Neozygites* in *Entomophthoromycotina*, and several genera in *Zoopagomycotina*), numerous isolates of several other species have been sequenced (Figure 1). This discrepancy can be largely explained by practical interests since focal species are sometimes chosen for their relevance to human health or biotechnology [18]. In addition, some taxa are common in culture collections whereas other taxa are challenging to grow in vitro or presumably uncultivable [4,11]. The goal of this review is to (1) provide background into the methods that have been used to sequence zygomycete genomes and which genomes have been sequenced, and (2) consider insights into the genomic architecture, gene content, and evolutionary history gained from these data.

## 2. Sequencing the Zygomycetes

Axenic cultures remain the preferred and most straightforward method of obtaining high-molecular-weight genomic DNA for sequencing projects [19]. Typically, the easier a species is to grow in culture, the easier it is to obtain pure, unfragmented DNA of high quality. In some cases, however, DNA from a single cell or even nuclei can be extracted and sequenced. This technique has been especially successful for the mycoparasites or fungal parasites of animals (e.g., *Piptocephalis*, *Stylopage*, and *Zoopage* [11,20,21]) or for resolving questions on genetic diversity of the nuclei in *Glomeromycotina*, such as *Rhizophagus* [22]. When neither culturing nor single-cell genomics is possible, fungal DNA can be extracted along with that of the host organism. The two genomes can be distinguished bioinformatically (e.g., in the case of *Entomopthorales* pathogens of insects [23]). In rare cases, high-quality tissue from field-collected fruiting bodies can be used to generate genomes (e.g., *Endogone*, *Jimgerdemannia*, and *Modicella* [12,14,24]).

The diversity of DNA sequencing technologies allows researchers to select the tools that are most appropriate for addressing specific questions but also requires setting minimal reporting standards for each type of sequencing technology. During sequence processing, genomic DNA is cut and sequenced in length spans from 100–300 base pairs (short reads, Illumina) to 1000–60,000 base pairs (long reads, PacBio, or MinION). Illumina short-read sequencing remains the dominant technology used to generate fungal genomes, while long-read technologies such as PacBio and Oxford nanopore are being used with increased frequency, especially for larger genomes (Appendix A). Illumina sequencing generally provides the greatest sequencing quality and accuracy, although millions of short raw reads (150–300 bp) are produced; as a result, genome assembly can be challenging especially with genomes featuring large amounts of repetitive DNA. Assembly quality might be adjusted by combining the use of short reads from Illumina with long reads from another sequencing platform. The use of longer reads (up to 60,000 bp) facilitates read alignment and/or mapping, often achieving the length of the chromosome [25] and allows precise resolution of repetitive sequences of rDNA [26]. However, PacBio and other long-read technologies typically demand higher quality and quantity of DNA than Illumina. On the other hand, these technologies are changing rapidly as different companies compete for the market share and make efforts to increase the number and length of reads and to maintain and improve sequence accuracy [27].

The majority of zygomycete genomes are assembled de novo, even when assemblies of closely related species are already available. Choice of assembly software usually depends on the data type. For example, AllPaths-LG works best for assembling Illumina short-read sequences, while Falcon is preferred for longer PacBio reads (Appendix A) [28]. The most widely used genome assemblers for zygomycetes include gsAssembler (Roche Diagnostics, Indianapolis, IN, USA), Newbler [18], Velvet [29], Arachne, Celera [30], HGAP [31], JGI Jazz (especially among earlier assemblies from 2001–2014) [32], Ray [33], CLC Genomics Workbench 20.0 (https://digitalinsights.qiagen.com, accessed on 10 July 2023), MaSuRCA [34], Falcon, AllPaths-LG [35], SOAPdenovo [36], SPAdes [37], SOLiD [38], Canu [39], FinisherSC [40], Flye [41], and MEGAHIT [42] (Figure 2).

Most zygomycete genome assemblies are similar in size to the average fungal genome: 30–50 Mb (Appendix A). Notable exceptions include species in *Entomophthoromycotina*, *Glomeromycotina*, and *Mucoromycotina*, whose large genomes compare in size to *Arabidopsis thaliana* (135 Mb), *Caenorhabditis elegans* (100 Mb), or *Drosophila melanogaster* (140 Mb) [43,44]. However, there are still far smaller than the genomes of some of the Basidiomycete fungi that cause rust diseases on plants, which includes the largest fungal genome sequenced to date (*Austropuccinia psidii*, 1 Gb) [45]. Most zygomycete genomes contain over 10,000 predicted genes. *Rhizophagus* spp. (*Glomeromycotina*) are a notable exception; they contain more than double the typical number with 26–30 K predicted genes (Figure 3). The smallest numbers of genes are found in *Zoopagomycotina* (4–8 K), followed by 6–11 K in *Kickxellomycotina*, 9–16 K in *Entomophthoromycotina*, *Mucoromycotina* and *Mortierellomycotina*, and 20–30 K in *Glomeromycotina* (Appendix A). However, these numbers may change for *Entomophthoromycotina* and *Zoopagomycotina* since there are only a few representative genomes available for those lineages.

During the assembly process DNA sequences are combined into contigs, which are sometimes merged to form scaffolds (if sufficient evidence exists to support these merges). These scaffolds of assembled DNA sequences should ideally equal the number of chromosomes [46]. In practice, the number of scaffolds rarely matches exactly the number of chromosomes and, for most species, the exact number of chromosomes is unknown [47]. Many genomes of easily culturable *Mucoromycota* with small genomes have the lowest number of scaffolds (e.g., 12 in *Actinomortierella wolfii*), while the genomes of species with larger genomes (e.g., taxa in *Glomeromycotina* and *Entomophthoromycotina*) are more fragmented such as 210 contigs in *Rhizophagus irregularis* [26], or 373,021 in *Massospora cicadina* [48]. These larger assemblies are due to larger overall genomes (e.g., larger number of chromosomes) and because of the difficulties with assembly caused by polyploidy, numerous repeats, and transposons [49]; these aspects are prevalent in many *Mucoromycota* and *Zoopagomycota* genomes (Appendix A). However, the sequenced genomes with hundreds of thousands of scaffolds often reflect low sequence coverage relative to the size of the genome, resulting in highly fragmented assemblies. All these factors contribute to the number of the scaffolds, not reflecting the actual numbers of chromosomes.

Genome quality is also improved by increasing the sequence coverage. Some zygomycete genomes have been sequenced at a depth of 500–700× and many genome projects aim for a mean depth of ca. 100× coverage (Appendix A). High-quality genome assemblies rely on increased sequencing depth, particularly in cases of high intraspecific polymorphism or variable genome architecture, e.g., species of *Glomeromycotina* and many noncoding regions and transposons, as in the genomes of *Entomophthoromycotina* species. However, some researchers have used low-coverage genome sequencing (ca. 10× coverage) as a cheap and efficient method to obtain a smaller number of orthologous gene regions for phylogenomic analyses [12,15,24].

## 3. Genomes and the Evolution of the Zygomycetes

The first genome sequence of a zygomycete fungus was from a clinical strain of *Rhizopus delemar* that was isolated from a patient with mucormycosis—a highly destructive and lethal infection that is typically seen in immunocompromised hosts. Sequencing of the *R. delemar* genome was driven from the clinical perspective, but it also provided the first genome of a fungus outside of the *Dikarya* [18]. Moving forward a decade, a landmark project generated genomic sequences of all species known to cause mucoromycosis in humans. Thirty zygomycete genomes were sequenced and assembled for this project [50]. Another substantial advance in zygomycete genomics was made by the ZyGoLife team, comprising the efforts of 12 mycological labs across the USA and Canada in collaboration with the US Department of Energy Joint Genome Institute (JGI) [14]. The phylogenomic study by Spatafora et al. [6] found that the former *Zygomycota* phylum was not monophyletic but instead comprised two distinct lineages—the primarily plant-associated saprotrophic and mycorrhizal *Mucoromycota* versus the primarily mycoparasitic or animal-associated parasites and commensals in the *Zoopagomycota*. However, further analyses using phylogenomics data to resolve the evolutionary relationships of zygomycetes have yielded slightly conflicting results. Although *Zoopagomycota* and *Mucoromycota* are maintained as monophyletic lineages, a recent study by Li et al. [51] found that *Zoopagomycota* and *Mucoromycota* formed one larger monophyletic clade in some (but not all) of their analyses. In contrast, other recent phylogenomic studies by Strassert and Monaghan [52] and Galindo et al. [53] recovered a topology that was similar to Spatafora et al. [6], where the *Zoopagomycota* were resolved as the sister group to all terrestrial fungi. In many of the studies, the position of *Glomeromycotina* has been problematic, suggesting that more work will be needed to resolve the placement of this clade [51,54].

One of the greatest debates in early-diverging fungal evolution was the ploidy of the dominant life form given the predominantly haploid nature seen in the *Dikarya*. One outcome of the analysis of genome sequences of zygomycetes species is that diploids are more common than expected [9,55,56,57].

### 3.1. Mucoromycota

*Mucoromycota* comprises many critically important plant symbionts, including the arbuscular mycorrhizal fungi (*Glomeromycotina*) [58], *Endogonales* (*Mucoromycotina*) [24], “fine root endophytes” (*Mucoromycotina*) [59], and putative root endophytic species including *Mortierella* (*Mortierellomycotina*) [60], *Pygmaeomycetaceae* (*Mucoromycotina*) [61], and *Umbelopsis* (*Mucoromycotina*) [62]. On the other hand, many fungi within this group are also noted plant pathogens (e.g., *Rhizopus microsporus*, which causes rice seedling blight, and various *Choanephora*, *Gilbertella*, and *Mucor* species that cause post-harvest rots) as well as numerous species (*Actinomucor*, *Apophysomyces*, *Cokeromyces*, *Cunninghamella*, *Lichtheimia*, *Mucor*, *Rhizomucor*, *Rhizopus*, *Saksenea*, *Syncephalastrum*) that can cause the disease mucormycosis in humans and other animals [63].

Most zygomycete genomes are of species within *Mucoromycota* (Figure 1); as a result, much of what we know about genome evolution among early-diverging fungi comes from this phylum. Genome duplication appears to have played a significant role in their evolution, as first described for *Rhizopus delemar* and *Mucor circinelloides* [18,64]. Genome sequencing has also revealed a diversity of endobacteria that may be found as symbionts in the *Mucoromycota*, e.g., in *Rhizopus microsporus* and *Endogone pisiformis* (*Mucoromycotina*), *Linnemannia elongata* and *Mortierella alpina* (*Mortierellomycotina*), and *Rhizophagus irregularis* (*Glomeromycotina*) [65,66,67,68,69,70]. Many plant-associated fungal lineages contain endobacteria, but the potential role of these symbionts in the land colonization process remain to be characterized. “Foreign” DNA in the form of diverse mycoviruses has also been frequently detected during genome sequencing of zygomycetes [30,71]. Genomics enabled the systematic study of genome architecture [72,73,74], genetic manipulations [75], and intra-isolate genomic variations [76]. Genome-wide searches and comparisons have also detected evidence for transposon spread through genomes [77,78,79] and uniquely high 6-methyladenine DNA modifications in zygomycete fungi as compared to *Dikarya* [80]. Studies on metabolic patterns across the fungal kingdom, the search for the mating gene loci and protein families [81,82,83], and the phylogenies of certain taxonomic groups [84,85] have also been conducted using-whole genome sequencing data. Genome-based research has also provided insights into the trophic modes of zygomycete fungi, including the evolution of mycorrhizae [24,86] and commonalities in the pathogenicity of Mucoralean fungi [50,87].

### 3.2. Mucoromycotina

Genome sequencing of this subphylum has been the most comprehensive among all the basal fungal lineages although genomes are still unavailable for several genera, including species of *Siepmannia*, *Chlamydoabsidia*, and *Utharomyces*, as well as for two new root-associated species of *Pygmaeomyces* of the new genus and family *Pygmaeomycetaceae* [61]. Furthermore, the plant-associated order *Endogonales* remains highly undersampled despite recent advances [24,88]. Some of its “fine root endophytes” colonize early-diverging plants, and sometimes co-occur with AMF [89,90]. The average genome size in *Mucoromycotina* species ranges from 35 to 50 Mbp, with species of *Umbelopsis* at the low end of the spectrum (22–30 Mbp) and *Endogonales* such as *Jimgerdemannia* species at the higher end (240 Mbp) (Appendix A). For some genera, several hundred genomes have already been sequenced. For example, there are >220 genomes available for species of *Rhizopus*, mostly of *R. arrhizus* (syn. *R. oryzae*) and its varieties and synonyms, and of *R. microsporus*, *R. delemar*, and *R. stolonifer* (Appendix A). *Rhizopus* is increasingly studied as a model organism thanks to its availability in culture collections and to its important role in biotechnology and human health. Genomic information might be helpful for future studies of pathogenicity in *Rhizopus* species since infections by members of this genus are the leading cause of human mucormycosis [50,84]. Phylogenomic data suggest that species of *Rhizopus* are part of a well resolved monophyletic group. However, this is in stark contrast to the situation among members of the genus *Mucor* (>150 genomes), which is incredibly diverse and is likely polyphyletic (Figure 4). The additional *Mucoromycotina* genomes consist mostly of saprotrophic species that are relevant to human health (species of *Lichtheimia* and *Cunninghamella*) and biotechnology (*Phycomyces blakesleeanus*) [91,92,93]. Genomes of *Endogonales* (as well as *Glomeromycotina*) contain effectors to facilitate plant–fungus interactions, which apparently aided in the terrestrialization of Earth by early land plants. Genome information helped to describe and clarify the phylogenetic placement of new lineages and species, such as *Bifiguratus* [94] and *Calcarisporiella* [95].

### 3.3. Mortierellomycotina

This subphylum contains an estimated 170 species [96] and was recently circumscribed into 13 monophyletic genera using multi-locus and low-coverage genome sequencing [12]. Reference genome sequences of both low quality and high quality are now available for representatives for most genera, including *Actinomortierella*, *Benniella*, *Dissophora*, *Entomortierella*, *Gamsiella*, *Gryganskiella*, *Linnemannia*, *Lobosporangium*, *Lunasporangiospora*, *Mortierella*, *Necromortierella*, *Podila*, and the sporocarpic genus *Modicella* [12,97]. A high diversity of *Mortierellomycotina* species can be isolated from soil, but it appears that genera may differ in their ecological roles. For example, *Necromortierella* species appear to be necrotrophic, whereas *Actinomortierella* spp. are often associated with millipedes, and *Entomortierella* spp. are often associated with arthropods [98]. Many genera, such as *Linnemannia* and *Podila*, have been recovered as rhizosphere associates and endophytes of plant roots [99,100], and they appear to improve plant growth and impact flower and seed production [101]. Several species such as *Mortierella alpina* have been sequenced due to their use in industry and biotechnology for lipid production [102].

### 3.4. Glomeromycotina

Arbuscular mycorrhizal fungi (AMFs) are arguably the most common and oldest symbionts of terrestrial plants [58], credited with the role in the facilitation of plant transition to the terrestrial habitat [103]. AMFs supply their plant hosts with mineral nutrients translocated from the soil in return for plant-assimilated carbon [104]. Yet, there are not many genomic sequences of these organisms available due to historical impediments hampering sequencing projects, including multinucleate spores with different genotypes, obligate biotrophy of AMFs, and the initial uncertainty concerning their genome sizes, with conflicting size estimates in *Rhizophagus irregularis* ranging from 14.1 [105] to 154.8 Mb [106]. Once these obstacles were surmounted, genomic data confirmed that AMFs have unusually large genomes, ranging from 153 Mb in *R. irregularis* [26] to 773.1 Mb in *Gigaspora margarita* [107]. This size enhancement relative to other *Mucoromycota* is driven by lineage-specific expansions of gene families and massive proliferation of transposable elements [26,86,107]. Genomic data also revealed that AMFs display several other features that differentiate them not only from *Mucoromycota* but also from other fungi. For example, despite the lack of morphological evidence of sexual reproduction, the *R. irregularis* genome harbors a candidate mating-type locus with two genes encoding homeodomain-like transcription factors [108]. If these homeodomain-like genes indeed control sex, then regulation of the reproductive processes in AMFs would resemble mating in *Dikarya* rather than in *Mucoromycota*, which rely on HMG domain proteins for sex determination [109]. *Rhizophagus irregularis* differs also from other fungi in the organization of its rRNA operons. Instead of being tandemly arrayed as they are in other fungi, the rRNA operons in *Glomeromycotina* are dispersed individually across the genome [26]. This important discovery put an end to a long-lasting debate on whether intraindividual rRNA gene variation typical for AMFs is distributed among dissimilar nuclei [110] or contained in each nucleus [111], supporting the latter scenario. Lastly, genome-based inferences of metabolic capacity revealed that AMFs lack the gene encoding fatty acid synthase [112,113,114], which makes them obligately dependent on plant hosts for lipids [115,116,117,118]. As in other fastidious organisms, this information is important for devising cultivation strategies that complement fatty acid auxotrophy of AMF [119,120]. Even though accumulation of genomic data for AMF has lagged behind other *Mucoromycota*, the insights gathered so far have been critical for resolving several puzzles in the biology of these unique organisms. PacBio HiFi and Hi-C sequencing of all available AMF heterokaryons helped to discover the two sets of coexisting homologous chromosomes in one heterokaryon. Genes required for plant colonization were found in gene-sparse, repeat-reach compartments [121]. Another important question that remains to be addressed is which genomic features of AMFs contribute to superior yield outcomes in agronomic crop species, as AMFs do not universally benefit all plant hosts.

### 3.5. Zoopagomycota

In contrast to *Mucoromycota*, most species of *Zoopagomycota* are parasites or pathogenes of animals and fungi. This includes the insect pathogenic group *Entomophthoromycotina* (some members of which exhibit behavioral control over their hosts) [122], the *Kickxellomycotina* that includes a mixture of ecologies (including commensalistic arthropods gut fungi) [15], and *Zoopagomycotina*, which are parasites of fungi (mainly in *Mucoromycota*) and microinvertebrates (e.g., amoebae, nematodes, rotifers) [20]. There are also opportunistic human pathogens in this phylum, including species of *Basidiobolus* and *Conidiobolus* [123]. Axenic culturing of some fungi in *Zoopagomycota* is possible [124,125] but extracting high-quality genomic DNA from many species remains difficult. Sometimes there is a need to extract genomic material of the parasite together with the host (often another fungus or insect) and separate their genomes during assembly, which creates additional challenges. Small amounts of available DNA have necessitated the use of single-cell DNA extraction in some cases, especially in species with dramatically reduced vegetative mycelial structures. Much greater success has been achieved for putative saprotrophic fungi such as dung-inhabiting species of *Kickxellales* (Appendix A). Genomes and transcriptomes have been used primarily to resolve various evolutionary questions, including the monophyletic origin of *Zoopagales* [20], evolution of ploidy from a diploid ancestral stage [11], and loss of a large number of pectinases and other plant cell-wall-degrading enzymes in *Zoopagomycota* [126]. Genome data were also successfully used to resolve evolutionary relationships within particular taxonomic groups, such as *Massospora* [98], to track other evolutionary events such as horizontal gene transfer events in gut fungi [127], and to reveal the genetic toolbox in insect symbionts [128]. These studies help us to understand the evolutionary trajectories of genome size, namely, a dramatic increase in genome size for the insect pathogens and insect endosymbionts as compared to the genome size in the saprobes and mycoparasites. Genome data have also helped to clarify fundamental differences among *Zoopagomycota* lineages in their production of secondary metabolites [129], as well as metabolic pathways in general [11]. Mating genetics in this phylum is still not well understood, and that is a question that remains to be addressed with genomic data.

### 3.6. Kickxellomycotina

This subphylum contains an interesting mix of mycoparasites (*Dimargaritales*), putative saprotrophs (*Kickxellales*), and obligate insect endobionts (*Asselariales*, *Barbatosporales*, *Harpellales*, and *Orphellales*). The *Dimargaritales* are similar to mycoparasites in the *Zoopagales* in that they form penetration hyphae (haustoria) and mostly attack similar host species in the *Mucoromycota*. In contrast to the *Zoopagales* mycoparasites, species of *Dimargaritales* appear to be rare in the environment and have mostly been isolated from dung rather than soil. Recent genomic analyses have also highlighted that members of the *Dimargaritales* that have larger genomes and more predicted secondary metabolite genes compared to their close relatives [15,129]. Furthermore, single-cell genome data have suggested that *Dimargaris cristalligena* may be nonhaploid and missing certain enzymes from metabolic pathways (e.g., thiamine biosynthesis and biotin metabolism) [11]. Identifying such genomic features of symbiotic species can provide targets for further experimental tests to understand the molecular interactions between hosts and parasites.

On the other hand, the *Kickxellales* is the only group of *Zoopagomycota* containing mostly saprotrophic species, and many appear to be rare in the environment, having only been reported in the literature one or a few times. An exception is the genus *Coemansia*, the species of which are commonly isolated from soil and dung samples [4,130]. Genome sequencing efforts have demonstrated that *Kickxellales* species have small genomes (less than 40 MB) and variability in their predicted secondary metabolite genes, with *Coemansia* and *Linderina*, along with more polyketide synthases than other *Kickxellales* and *D. cristalligena* having more nonribosomal peptide synthetases than other *Kickxellales* [129]. A preliminary rDNA phylogeny showed polyphyly among *Kickxellales* [130], and a large-scale sequencing project that utilized low-coverage genome data from >100 isolates to test these hypotheses, as well as explore the diversity of *Coemansia* species, resolved many of these relationships [15].

The gut fungi (*Trichomycetes*) include arthropod symbionts for which several draft genomes have been available since 2016 [131]. These fungi are distinct from the saprotrophic and coprophilic relatives (i.e., *Kickxellales*) and belong to four distinct phylogenetic lineages—*Asellariales*, Barbatosporales, *Harpellales*, and *Orphellales* [132]. *Asellariales* are associated with isopods and spring tails (*Collembola*) [133]. Species within the three genera (*Asellaria*, *Baltomyces*, and *Orchesellaria*) have been recovered from the both terrestrial and aquatic hosts [133], but their phylogenetic relationships remain unclear, as sequence data have only been obtained from *Asellaria* [132], and no cultures are available for this group.

*Harpellales* species are usually found in lower *Diptera* insect larvae and have coevolved with their hosts for over 200 million years [134]. Species of *Orphellales* are stonefly nymph symbionts with unusual morphological characters (i.e., asexual and sexual spore shapes, and thalli protruding beyond the anus of insect hosts) [135], which cluster together with other *Trichomycetes* and the *Spiromycetales* for a sister clade to *Kickxellales* [15]. *Smittium* is a well-studied genus within *Harpellales* with approximately 100 described species [136]. However, there are only nine whole-genome sequences available for these insect gut-dwelling fungi, and all were made from culturable *Harpellales* species, with the best assembly—*Capniomyces stellatus*—having 72 scaffolds [131]. *Barbatosporales* are monotypic, and the single species (*Barbatospora ambicaudata*) has only been recorded from black fly larvae collected in the Great Smoky Mountains National Park in Tennessee, obtained in axenic culture [137].

Although we only have a handful of *Harpellales* genomes, they have already expanded our knowledge and understanding of these enigmatic microbial fungi in multiple ways. For example, a mosquito-like polyubiquitin gene has been identified in *Zancudomyces culisetae* (*Harpellales*), which was presumably acquired via a horizontal gene transfer event [127]. Comparative genomics revealed a fungus–insect symbiotic core gene toolbox (FISCoG) that is shared among the insect-associated fungi (e.g., *Harpellales*, *Basidiobolus*, and *Conidiobolus*) and higher-ranked *Hypocreales* (*Ascomycota*). However, the insect pathogens (*Hypocreales* and *Entomophthoromycotina* members) tend to have genomes enriched in genes that are useful for pathogenic processes such as the platelet-activating factor acetyl-hydrolase coding genes, whereas the gut commensals have genomes enriched in cell adhesion genes for a successful gut-dwelling lifestyle [128]. In addition, *Harpellales* genomes also facilitated a kingdom-wide study to confirm the production of selenoproteins in early-diverging fungal lineages [138]. As a major group of early-diverging fungi, representing seven of the nine fungal species that utilize selenoproteins, *Harpellales* may take the advantages of selenocysteine over cysteine for specific oxidoreductase functions.

Unfortunately, no efforts have been successful in obtaining genome information of the *Asellariales*, *Barbatosporales*, and *Orphellales*, which colonize different host types. However, multigene analyses incorporating low-coverage genome data (along with data from *Asellariales* and *Orphellales*) suggested an alternate hypothesis for the evolution of fungal–insect associations. Contrary to previous topologies that suggested multiple events [132], the larger dataset showed a possible single origin of gut fungi [15]. Further evolutionary evidence may be identified with the help of their genome sequences to understand how these gut-dwelling microbial fungi interact with various aquatic insect hosts and evaluate the origins of the symbiosis.

### 3.7. Zoopagomycotina

*Zoopagomycotina* species are fascinating, partially due to their cryptic ecological niches and ability to parasitize various hosts such as amoebae, nematodes, rotifers, and other fungi. It is still challenging to culture most of the *Zoopagomycotina* isolates. Thus, it is not surprising to see that the *Zoopagomycotina* has the lowest number of available genomes, the qualities of which are not comparable to other lineages, especially the well-sequenced *Mucoromycotina*. Recently developed single-cell genomic techniques have provided an alternative, culture-independent method to obtain their genomic information [11]. On the basis of single-cell or multiple-cell libraries, the completeness of genome assemblies can be as high as 89.6% (*Piptocephalis tieghemiana* RSA 1565) [20]. The assembled *Zoopagomycotina* genomes indicate that this clade of fungi is characterized by small genome size, but the numbers of protein-coding genes are not necessarily low. Some of the genomes can encode close to 10,000 genes (e.g., *Zoophagus insidians*, a predator of rotifers and nematodes (Appendix A). It seems likely that these completely microscopic fungi are capable of more complicated and yet-to-be identified interactions with their animal and/or fungal hosts, but more work will be needed to explore these interactions.

Two additional *Zoopagales* have remarkably small genomes: a species of *Acaulopage* (11 Mbp) and a species of *Zoopage* (14 Mbp). No genomes have yet been published for the following genera: *Amoebophilus*, *Aplectosoma*, *Bdellospora*, *Brachymyces*, *Cystopage*, *Endocochlus*, *Euryancale*, *Helicocephalum*, *Reticulocephalis*, and *Sigmoideomyces*.

### 3.8. Entomophthoromycotina

This lineage remains the most insufficiently sequenced fungal group at the subphylum level. Despite many genomes from the genus *Conidiobolus*, there are only a few other species (all from the key family *Entomophthoraceae*) for which genomes are available. The genus *Conidiobolus* is a polyphyletic assemblage of multiple lineages. This group of *Conidiobolus*-like fungi was recently phylogenetically divided into three new families, using a combined multi-gene and phylogenomic approach, revealing that the ballistic conidia arose prior to their transition to a parasitic lifestyle [85,139]. Difficulties with culturing and, therefore, obtaining enough high-molecular-weight genomic DNA represent a reason why some species only have transcriptomes available. Alternatively, in some cases, the fungal pathogen is sequenced together with its host, and then the reads are separated bioinformatically. Aside from the polyphyletic genus *Conidiobolus* and entomophthoroid-like *Basidiobolus*, genome or transcriptome data are available only for a few species of *Entomophaga*, *Entomophthora*, *Massospora*, *Pandora*, and *Zoophthora*. No genomes yet exist for *Ancylistes*, *Apterivorax*, *Batkoa*, *Erynia*, *Eryniopsis*, *Furia*, *Macrobiotophthora*, *Meristacrum*, *Neozygites*, *Orthomyces*, *Tabanomyces*, *Thaxterosporium*, *Schizangiella*, and *Strongwellsea*. The largest genomes recovered are over 2000 Mbp in *Massospora* and 650 Mbp in *Zoophthora*, with possibly similar genome sizes in *Entomophthora* and *Entomophaga* (Appendix A). Phylogenetic analysis recently demonstrated that *Basidiobolus*, a genus of amphibian gut-dwelling fungi, does not belong to *Entomophthorales*, where it was traditionally placed, mostly on the basis of its production of forcibly discharged conidia that are morphologically similar to those in other entomophthoralean fungi [140]. The genus was difficult to place with rDNA data [141], but was inferred either as a separate clade sister to the *Mucoromycota* [51] or as the earliest diverging lineage within *Zoopagomycota* [128], on the basis of phylogenomic data. The ballistic spore dispersal mechanism, which differs structurally from that found in the *Entomophthorales*, is likely an example of convergent evolution among fungi that appear to be similar but are distantly related. In addition to being evolutionarily enigmatic, the genomes of *B. meristosporus* and *B. heterosporus* are larger than most other basal fungi and contained a much higher proportion of secondary metabolite-encoding genes than other *Zoopagomycota* species that have been studied thus far. Furthermore, *Basidiobolus* genomes contain numerous genes that putatively originated from horizontal gene transfer events from bacteria [129].

## 4. Insights from Genomics

### 4.1. Adaptation to Terrestrial Life Inferred from Genomes

The transition to a terrestrial environment placed novel constraints on species of early-diverging fungi, especially in terms of their resistance to desiccation and solar radiation. Genomes have been essential to studying which zygomycete genes made this transition possible.

Sensing of and response to light have been explored in the model *Mucorales* species *Phycomyces blakesleeanus*. Mutants impaired in photoresponses, especially phototropism, were isolated as reported in the 1970s [142]. However, the first genes related to photoresponses were not reported until 2006 [143]. These were identified using a candidate gene approach in which *P. blakesleeanus* was assumed to have similar blue light sensors as the *Dikarya* species [143,144], wherein a two-protein complex was first identified in *Neurospora crassa*, defining a sensing and signaling system that uses a light-sensing domain coupled to transcription factor roles for signal transmission. With the release of the *P. blakesleeanus* genome sequence in 2010, identifying a candidate gene for *madB* as encoding the interacting protein was achieved using BLAST against the genome sequence [145]. The genome sequence was performed along with that of *M. circinelloidies* and revealed an expansion of this pair of genes required for light responses (3–4 copies of each). Being able to make targeted gene mutations in *M. circinelloides* of these homologs, as based of genome sequence information, provided a clear example of subfunctionalization after gene duplication [146]. The identification of the mutations in *madI* and *madC* mutants in *P. blakesleeanus* used traditional genetic mapping with molecular markers derived from the comparison of the genomes between two strains. With the regions narrowed, candidate genes could be identified and confirmed by sequencing [64,147]. These genes play conserved roles in photobiology in the *Dikarya* species, suggesting that this innovation for terrestrial life had already evolved in early-diverging fungal lineages. For instance, phenotypic analysis of the *P. blakesleeanus madA* and *madB* mutants indicated that this pair of genes has a conserved function in protecting fungi from harmful ultraviolet light [148].

In addition to responding to light, many *Mucorales* species can sense gravity. This has been correlated with the formation of vacuolar crystals in *P. blakesleeanus* [149]. Through purification of such crystals, followed by mass spectrometry of the associated proteins using the mass database derived from the genome sequence, the OCTIN protein was identified. A gravity-insensitive mutant had a mutation within the corresponding gene. Remarkably, phylogenetic comparison revealed that the origin of the gravity-sensing gene OCTIN in the *Mucorales* was most likely via a horizontal gene transfer from a bacterial species [150]. Other challenges associated with terrestrial life, such as fluctuations in temperature and desiccation, have yet to explored from a genomic standpoint among zygomycetes. Some of the early diverging lineages such as *Glomeromycotina* and *Mortierellomycotina* lack ergosterol, what can increase their membrane stiffness [77,112]. Ergosterol has long been considered as one of the regulators of dry–wet cycles [151].

### 4.2. Endobacteria

Symbiotic associations with highly coevolved endocellular bacteria stand out among the most prominent features differentiating *Mucoromycota* from other fungi, including the *Dikarya* [152,153,154]. Thus far, most of these endosymbionts have been identified as representatives of two bacterial families, *Burkholderiaceae* and *Mycoplasmataceae*. While some of the *Mucoromycota* associations with bacteria are mutually beneficial for both partners, in others, the fungi appear to be negatively impacted by their endosymbionts. The application of “omics” tools has been essential in elucidating the molecular underpinnings of the mutualism between *Glomeromycotina* and their ancient and uncultivable endosymbiont “*Candidatus* Glomeribacter gigasporarum” (CaGg, *Burkholderiaceae*) [155]. In this symbiosis, CaGg improves hyphal proliferation in the presymbiotic growth stage of the host fungus [156], which can be attributed to endobacteria-mediated reprograming of the fungal metabolism [157,158]. “Omics” approaches were also critical in understanding the biology and devising a cultivation strategy for *Mycoavidus cysteinexigens* (*Burkholderiaceae*) [159], a cysteine auxotroph and ancient antagonistic endosymbiont of *Mortierellomycotina* [69,160,161]. Comparative genomics also revealed that “*Candidatus* Mycoavidus necroximicus” produces several nematocidal metabolites that protect the fungal host *Mortierella verticillata* from feeding by nematodes [162]. Endobacteria have also been shown to regulate and control mating and asexual reproduction in *Rhizopus microsporus* [163], produce toxins (rhizoxin) that enable pathogenicity on plants, and provide protection from predators [164], as well as potentially protect fungal spores from the human immune system via anti-phagocyte activity [165]. The ease of partner manipulation in the mutualism between *R. microsporus* and *Mortierellaceae* (*Mucoromycotina*) and *Mycetohabitans* spp. (formerly *Burkholderia*) [166,167] elevated this association to a model system for studying fungal-bacterial symbioses [163,168].

While the symbioses of *Mucoromycota* with *Burkholderiaceae* endobacteria are well characterized for some species, *Mucoromycota* associations with *Mycoplasmataceae*, referred to as *Mollicutes*- or mycoplasma-related endobacteria (MREs) [169,170], are only beginning to be unraveled. MREs have been detected in all three subphyla of *Mucoromycota*: *Glomeromycotina* [169], *Mortierellomycotina* [171], and *Mucoromycotina* [24,172]. MREs of *Glomeromycotina* are classified as “*Candidatus* Moeniiplasma glomeromycotorum” (CaMg) [173]. Genomic features of CaMg have been examined in depth [66,174], including evidence of gene acquisition from fungal hosts. However, the role of CaMg in the biology of *Glomeromycotina* remains uncertain. Naito et al. [174] speculated that CaMg is an antagonist of *Glomeromycotina*. If this is confirmed, then CaMg would resemble the lifestyle of the MREs associated with *Mortierellomycotina* [171].

Moreover, bacterial–fungal interactions seem to be much more common and diverse than previously thought [175], and *Mucoromycota* representatives are no exception. Telagathoti et al. [176] showed recently that 30% of *Mortierellales* isolates were associated with different groups of bacteria. Another survey of *Mucoromycota* cultures from soil indicated diverse *Paraburkholderia*-related sequences detected with 20% frequency from *Mortierella* and *Umbelopsis* strains [154]. The ecological meaning of these interactions remains to be described.

Outside of the *Mucoromycota*, however, the importance or prevalence of endosymbiotic bacteria remains less clear. Despite bacterial sequences always present in the environmental samples of entomophthoralean fungi (A.P. Gryganskyi, personal observation), nothing is known about their connection to the entomopathogens and their role in pathogenic processes that impact insects. However, six recently constructed metagenome assemblies of four various bacterial species in connection with *Massospora cicadina* might indicate their symbiotic relationship to this entomopathogenic species [177]. Among *Zoopagomycotina*, Davis et al. [20] detected 16S sequences with close matches to *Mycoavidus cysteinexexigens* in the genome of *Stylopage hadra* (nematode parasitic fungus). A recent genomic assessment of *Kickxellomycotina* fungi did not reveal any obvious colonization by endosymbiotic bacteria [15], suggesting that these bacterial associations might be less important or prevalent in *Zoopagomycota* than in *Mucoromycota*.

### 4.3. Secretomes, Gene Clusters, and Secondary Metabolites

All fungi explore and exploit their environment by secreting various substances. The secretome composition of basal fungal lineages was shaped by diversification early in their evolution. Characterization of secretome compositions of 132 zygomycete genomes revealed that current trophic mode plays a less significant role as compared to the phylogenetic relationships of the fungi [178]. This is in part because *Mucoromycota* and *Zoopagomycota* each had their own unique lineage-specific expansion of secreted digestive enzymes, according to their historical substrate differences and their different evolutionary trajectories [178].

Genomic screenings of available genomes for novel natural compounds also show promising potential [179,180,181]. These screenings discovered the presence of gene clusters harboring polyketide synthases (PKS), non-ribosomal peptides, terpenoids, and L-tryptophan dimethylallyl transferases with transcription factor coding genes in proximity to some of the gene clusters. Since then, NRPS clusters have been detected in some *Mortierellomycotina* and *Mucoromycotina*. For example, the mycelium of *Mortierella alpina* produces diverse novel compounds [182]. Similarly, the transcripts corresponding to PKS and NRPS were recently described in transcriptomes of five *Mucor* species [183]. *Mucor circinelloides* was shown to produce carotenoids and terpenoids [180], and the genes associated with terpene biosynthesis were identified recently in *Mucor* and *Rhizopus* genomes [74]. Siderophore production has been reported from *Glomus* [184], *Apophysomyces* [185], *Rhizopus* [186], and *Basidiobolus* [129]. Rhizoferrin from *Rhizopus delemar* has been further characterized and showed a glycosylation pattern [187] that was classified as a member of the sodium (Na)–iodide symporter gene family [186]. The best-studied compound isolated from early diverging fungi is an antimitotic agent called rhizoxin produced by an endosymbiotic bacterium (*Mycetohabitans rhizoxinica*) and encoded by a huge NRPS–PKS cluster [166]. The genome of amphibian gut symbiont *Basidiobolus* is enriched with the genes and gene clusters involved in the production of secondary metabolites through horizontal gene transfer from bacteria [129]. Genomes of mycoparasitic *Dimargaritales* (*Kickxellomycotina*) are also enriched for secondary metabolites [15], especially when compared to mycoparasitic *Zoopagomycotina* (*Piptocephalis* and *Syncephalis*) whose genomes appear depauperate in these genes [129]. These groups of mycoparasites are very similar in their morphology and ecology; thus, the reasons underpinning the observed difference in metabolite genes remain unclear.

### 4.4. Transposable Elements

Transposable elements (TEs) are abundant in zygomycete genomes [79]. Genomes of *Entomophtorales*, *Basidiobolales* [129], and *Endogonales* [24] are particularly rich in TEs, while species of *Umbelopsidales* [188] and *Zoopagomycotina* [11] have very few. Genomes of *Rhizopus* species comprised approximately 40% repetitive sequences with only ~10% potentially active elements [84]. In taxa with more compact genomes such as *Umbelopsis isabellina*, transposons account for as little as 5% of their genome size. In contrast, *Jimgerdemannia flammicorona* can have more than 77% of its genome occupied by repetitive elements and transposons [24]. In most of the analyzed genomes, long terminal repeat retrotransposons from the Ty3/Gypsy family dominate the TE landscape [84]. *Mucoromycotina*, *Kickxellomycotina*, and *Blastocladiomycota* possess transposons with a tyrosine recombinase from a gene family which seem to be lost in *Dikarya* [78]. While the function of mobile elements in zygomycete genomes is not well understood, in Ascomycota, large mobile elements (termed “starships”) may carry a variety of genes (e.g., secondary metabolites, virulence factors, and heavy-metal tolerance) and contribute to the overall variation in genome structure [189,190].

## 5. Conclusions

There has been renewed research interest in the zygomycete fungi during the last decade, and this is reflected in the rapidly increasing number of genome sequences from this group of fungi. While the quality of the genomes varies according to the sequencing platform and starting materials available, the genomes are sufficient for phylogenomic reconstructions, as well as for the discovery of functional genes and gene clusters of interest. Much of the work has focused on several key topics, including the evolution and phylogeny of the group, adaptation, and transition to a terrestrial habitat (e.g., light and gravity sensors, and reinforcement of cell-wall structure to tolerate a nonaquatic environment), genome duplication, endosymbionts and horizontal gene transfer, genome architecture, pathogenicity, and coevolution with host species. The number of genomes has grown rapidly over the past decade, and we expect that the next decade will bring increased taxon sampling (e.g., new genomes for taxa that have not yet been sampled) and improvements in genome quality due to advances in long-read sequencing technologies and bioinformatics approaches. In the future, we expect that genomic analyses of these fungi will take new directions, including a detailed comparison of the genomes of aquatic vs. terrestrial lineages, structural studies, a comparison of the zygomycete vs. *Dikarya* genomes, experimental manipulations of host/parasite interactions to understand the molecular mechanisms underpinning pathogenesis, and building links among genomic, proteomic, and metabolomic data.

## Figures and Tables

**Figure 1 microorganisms-11-01830-f001:**
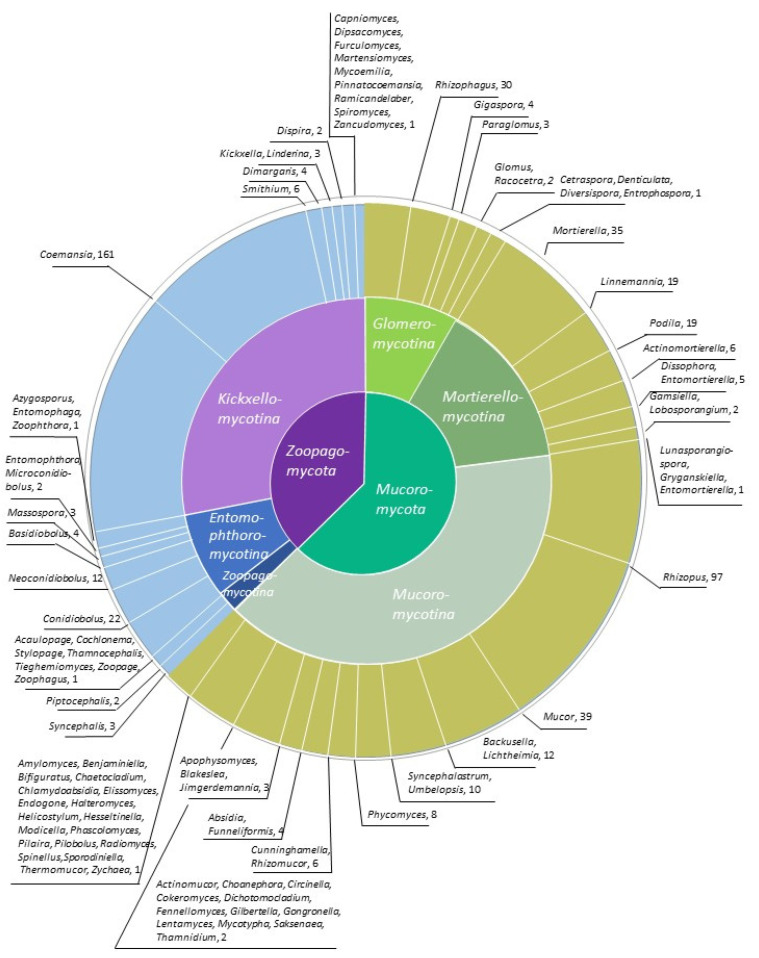
Distribution of sequenced genomes among genera. Different shades of green and brown—*Mucoromycota*; purple and blue—*Zoopagomycota*.

**Figure 2 microorganisms-11-01830-f002:**
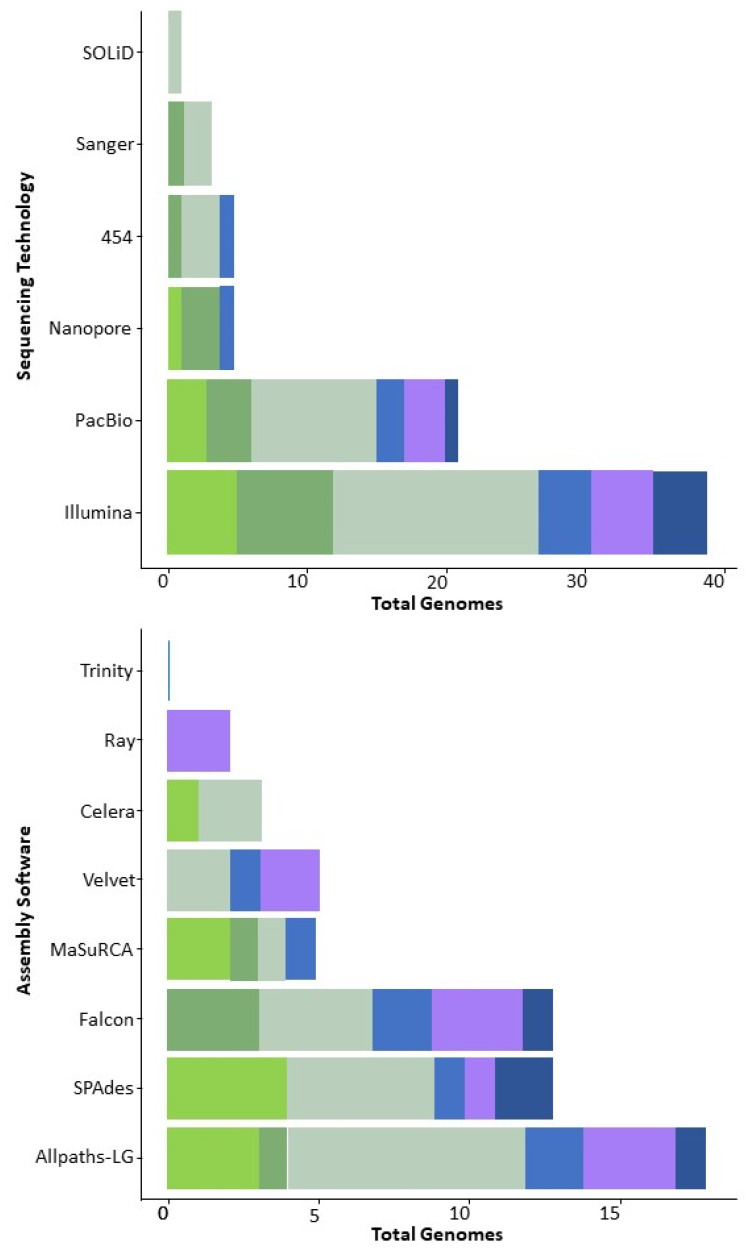
Approaches used in the generation of zygomycete genomes: sequencing (**upper panel**) and assembly (**lower panel**) technologies. Different shades of green and brown—*Mucoromycota*; purple and blue—*Zoopagomycota*.

**Figure 3 microorganisms-11-01830-f003:**
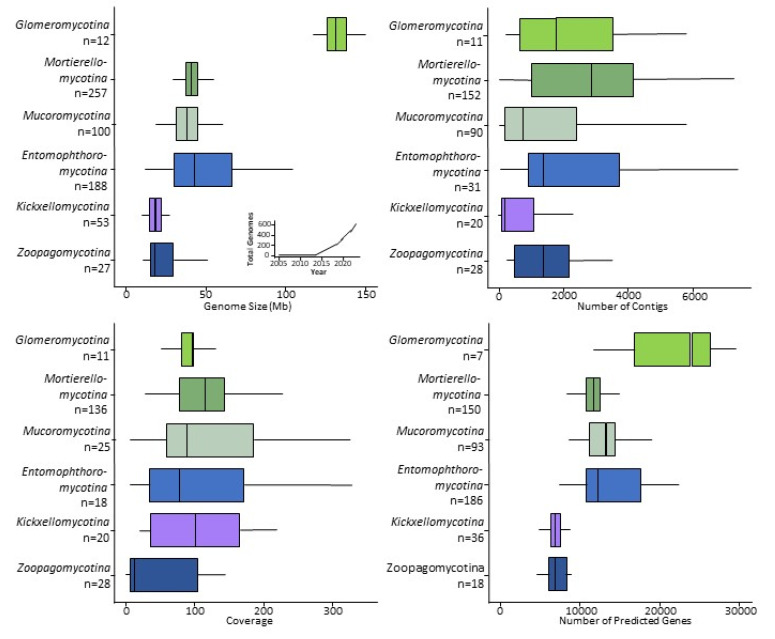
Genome statistics for zygomycete sequences. Sizes of the genomes and number of the genomes sequenced since 2005 (**upper left**), number of contigs in assemblies (**upper right**), estimated sequencing depth coverage relative to the single genome (**lower left**), and the number of the predicted genes (**lower right**). Different shades of green and brown—*Mucoromycota*; purple and blue—*Zoopagomycota*.

**Figure 4 microorganisms-11-01830-f004:**
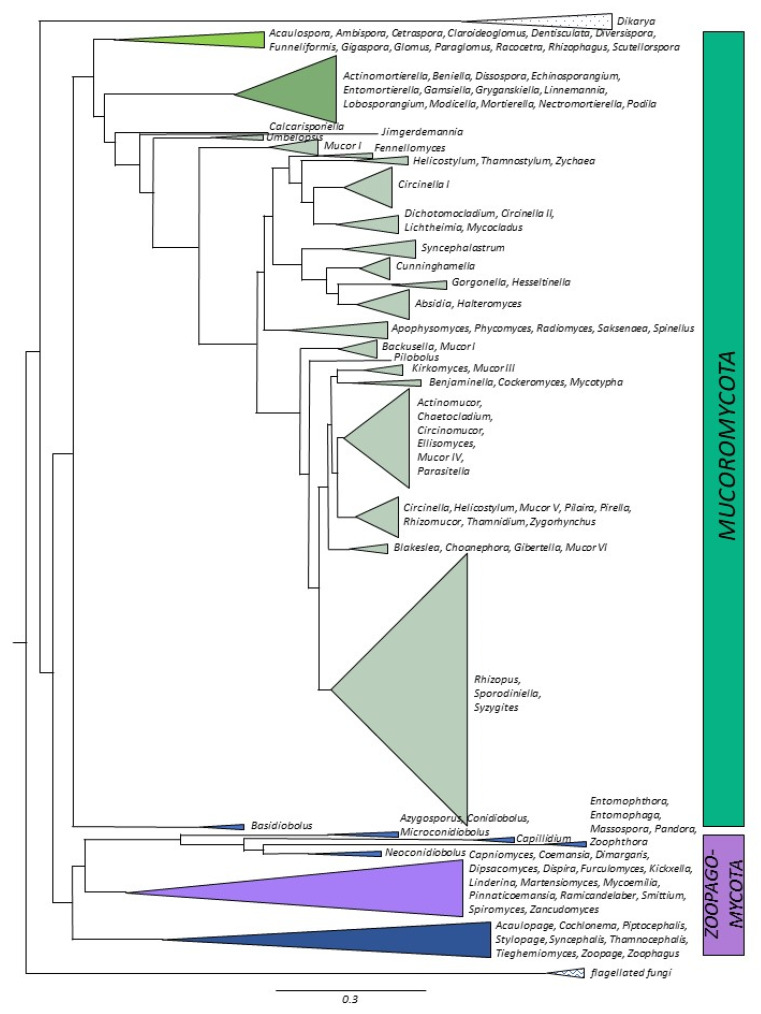
Distributions of the number of genomes sequenced imposed onto phylogenetic tree. Different shades of green and brown—*Mucoromycota*, purple and blue—*Zoopagomycota*.

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
