# Peer review of "Sequencing the Genomes of the First Terrestrial Fungal Lineages: What Have We Learned?"

_microorganisms, 2023, doi:10.3390/microorganisms11071830_

Round 1

Reviewer 1 Report

Article objectives are not clear, the writing should be clearer in the objectives. The novelty of the work should be better emphasized. The authors should elaborate more on their findings compared to other studies, to their importance.

Figures is not in printable quality. Also, some portions of the texts are losing their readability while sizing the image as per text area. Kindly provide better quality figure.

The discussion still needs improvement and explanation in detail. I recommend re-write and expanding: discussion section with updated literatures.

Please check the References in-text and end-list for uniformity in style.

The conclusion you have provided is quite brief and provides sufficient feedback on the main objectives of your study.

The authors are advised to improve the manuscript in terms of adequate language levels as well as research paper structure.

Author Response

Dear Reviewer,

Thank you very much for your review. Work on your comments improved the quality of our manuscript. Below are our responses to your review. We believe all these changes added more clarity to the text and made it easier to understand for the broader audience.

Figures is not in printable quality. Also, some portions of the texts are losing their readability while sizing the image as per text area. Kindly provide better quality figure.

We improved the figures and submitted the better-quality jpg files, also removed some errors.

Fig. 1 We increased the font of genera names and fixed the typos in Phycomyces, Umbelopsis, Linnemannia. Also, we added “Different tones of green-brown – Mucoromycota, purple-blue – Zoopagomycota” to the legend.

Fig. 2. We added “upper and lower panels” to the legend. falcon changed to Falcon.

Fig. 3. We added “upper left, upper right, lower left and lower right panels” to the legend, also removed the background of the n in Entomophthoromycotina, upper right pannel.

Fig. 4. Origin of the tree added, gray line below the figure removed, in the figure Backusella and Mucor I bold formatting removed.

The discussion still needs improvement and explanation in detail. I recommend re-write and expanding: discussion section with updated literatures.

We updated the literature through the text of the manuscript and standardized it in the literature list, below. Also, we added more contemporary literature sources: Asha & Vidyavathi 2009, Benny et al. 2016, Büttner et al. 2021, Davis et al. 2019, Espino-Vázquez et al. 2020, Ogura‑Tsujita et al. 2019, Partida-Martinez & Hertweck 2005, Reynolds et al. 2023, Richter et al., Rimington et al. 2019, Walsh et al. 2021, Wang et al. 2023.

We also added and updated relevant taxonomical and ecological information through the text of the manuscript, which added clarity. We standardized the use of “ and ‘ inverted commas, switched to double ones. We standardized the list of the authors affiliation, removed unnecessary zip codes and added Dept, School, etc. Also, we improved the writing.

Abstract

39 – “sequenced” added.

Introduction

79-81 We changed the last sentence of this paragraph to: “Hence, zygomycete genomes have and will continue to play an integral role in our current understanding of the initial transition of organisms to land [10].”

82 “in recent years” removed.

99 “to grow in vitro” added.

100-103 We also added the goal of our study: “The goal of this review is to 1) provide background into the methods that have been used to sequence zygomycete genomes and which genomes have been sequenced, and 2) consider insights into the genomic architecture, gene content, and evolutionary history gained from these data.”

Sequencing the zygomycetes

120 “recent proliferation” – changed to “diversity”.

133 “exciding” changed to “achieving”.

147 double mentioning of gsAssembler (Roche) removed.

182 “such as” – added to “210 Rhizophagus irregularis”, “and” changed to “or”.

189-190 “which in almost all sequenced fungal genomes is much larger than the true number of the chromosomes” changed to “not reflecting the actual numbers of chromosomes”.

191 700x changed to 700×

192 100x changed to 100×.

196 10x changed to 10×.

Genomes and the evolution of the zygomycetes

203 “useful” changed to “driven”

210 “Phylum Zygomycota” changed to “Zygomycota phylum”

223-226 We completely restructured the sentence as standing alone, now it is clearer: “One of the greatest debates in early diverging fungi evolution was the ploidy of the dominant life form given the predominantly haploid nature seen in the Dikarya.  One outcome of the analysis of genome sequences of zygomycetes species is that diploids are more common than had been expected [9,55–57].”

MUCOROMYCOTA

240 “role in mucoralean genome evolution” changed to “role in their evolution”

262 Pycnopodium removed as it was synonymized with Pilobolus recently.

263 new undersampled family of root associated fungi is added (Walsh et al. 2021).

265 We added “Some of its “fine root endophytes” colonize early diverging plants, and sometimes co-occur with AMF [82,83].”

Glomeromycotina

335 “has been lagging behind” changed to “has lagged behind”

336-337 We changed “One important question – coexisting two sets of homologous chromosomes in one heterokaryon with gene-sparse, repeat-rich compartments of the genes required for plant colonization, was resolved using PacBio HiFi and Hi-C sequencing of all available AMF heterokaryons” to “PacBio HiFi and Hi-C sequencing of all available AMF heterokaryons helped to discover the coexistence of two

ZOOPAGOMYCOTA

370 “and metabolic pathways in general” added.

370-372 We added “Mating genetics in this phylum is still not well understood, and that is a question that remains to be addressed with genomic data.”

Kickxellomycotina

394-398 We changed “A preliminary rDNA phylogeny showed polyphyly among Kickxellales [119], and a large-scale sequencing project is in progress that will utilize low coverage genome data from >100 isolates to test those hypotheses as well as explore the diversity of Coemansia species.” to “A preliminary rDNA phylogeny showed polyphyly among Kickxellales [121], and a large-scale sequencing project that utilized low coverage genome data from >100 isolates to test these hypotheses as well as explore the diversity of Coemansia species resolved many of these relationships [15].”

408 “Harpellales are” changed to “Harpellales species are”

409 “…with the hosts over 200 million years [114]. Orphellales are” changed to “with their hosts for over 200 million years [124]. Species of Orphellales are.”

436-439 We added: “However, multigene analyses incorporating low coverage genome data (along with data from Asellariales and Orphellales) suggested an alternate hypothesis for the evolution of fungal-insect associations. Contrary to previous topologies that suggested multiple events [123], the larger data set showed a possible single origin of gut fungi [15].”

442-443 We added “and evaluate the origins of the symbiosis.”

Zoopagomycotina

463 Added Reticulocephalis.

ENDOBACTERIA

521 “on endohyphal level” removed.

551-553 Added: “Comparative genomics also revealed that ‘Candidatus Mycoavidus necroximicus’ produces several nematocidal metabolites that protect the fungal host Mortierella verticillata from feeding by nematodes [151]”

554-557 Added: “produce toxins (rhizoxin) that enable pathogenicity on plants and provide protection from predators [153], as well as potentially protect fungal spores from the human immune system via anti-phagocyte activity [154].”

575-577 Added “Another survey of Mucoromycota cultures from soil indicates diverse Paraburkholderia-related sequences detected with 20% frequency from Mortierella and Umbelopsis strains [143].”

579 “Outside of Mucoromycota” changed to “Outside of the Mucoromycota

585-586 Added “Among Zoopagomycotina, Davis et al. [20] detected 16S sequences with close matches to Mycoavidus cysteinexexigens in the genome of Stylopage hadra (nematode parasitic fungus).”

SECRETOMES, GENE CLUSTERS & SECONDARY METABOLITES

597 “secretive” changed to “secreted”

618-623 Added: “Genomes of mycoparasitic Dimargaritales (Kickxellomycotina) are also enriched for secondary metabolites [15], especially when compared to mycoparasitic Zoopagomycotina (Piptocephalis and Syncephalis) whose genomes appear depauperate in these genes [120]. These groups of mycoparasites are very similar in their morphology and ecology, so the reasons underpinning the observed difference in metabolite genes remains unclear.”

TRANSPOSABLE ELEMENTS

633 added “the’ to “…from the Ty3/Gypsy…”

Supplementary Materials

660 Mucoromycota and Zoopagomycota changed to Italics.

Reviewer 2 Report

This is a timely and comprehensive review of the knowledge that has been gained from the genomes of the diverse organisms commonly referred to as zygomycetes. The review will be very useful in research and educational realms. Most of my comments are minor and have to do with style and presentation.

General comment: It would be useful to explain how the term zygomycete is being used in this manuscript.

Line 55. Add “a” before “more”

Line 55. “switching” and “more advantageous” are loaded words here. Why not acknowledge that certain lineages evolved to exploit new environments without making judgements about which lifestyles are best. There are still many flagellate fungi and unicellular fungi alive and well in aquatic and terrestrial environments.

Line 60. Delete “the” before “reproductive”

Lines 61-62. Does it follow that reproductive modes in the zygomycetes and lack of macroscopic fruiting bodies make these organisms more difficult to study than members of the dikarya?

Figure 2, top panel. I assume Nanodrop is supposed to be Nanopore.

Figure 2 legend and panels. The panels are not labeled A and B, but these designations occur in the legend. I suggest adding the labels to the figures or changing the legend to refer to upper and lower panels.

Figure 3. Same comment about letters as for Figure 2.

Figure 3 legend. Change “sequences zygomycetes” to “sequences of zygomycetes” or “zygomycete sequences”. Also, Delete “It covers” (its meaning is unclear and it is not necessary).

Line 162. Delete “the” before “contigs”.

Line 190. Change “the patient” to “a patient”

Line 202. Change “analysis” to “analyses”

Line 212. Perhaps change “turned to be” to “has involved”

Line 226. Change “that transposon” to “for transposon”

Line 310. Should “coexisting two sets” be changed to “two sets of coexisting”?

Line 311. Should the comma after colonization be replaced with a dash to make it complement the dash in line 310?

Line 401. It is not clear what the numbers in the following parenthetical expression refer to: “(e.g., 50 or 100)”

Line 416. A space is needed in “fromthe”.

Lines 457-464. This section seems to assume the reader will already know about the Phycomyces mad genes. I think it should be fleshed out a bit, and maybe the relationships these genes have to the wc genes of Neurospora and/or other members of the Dikarya should be clarified.

Line 508. The phrase “not only on endohyphal level” is not clear.

Lines 524-526. This sentence is not clear: “The secretome composition of basal fungal lineages is shaped when nutrition models diversified early in their evolution.”

Line 574. Perhaps change “expected” to “expect.”

The manuscript is well written but has a few places where wording or presentation can be improved. I have noted these in my comments to the authors.

Author Response

Dear Reviewer,

Thank you so much for very descriptive and helpful analysis of the text. Our responses to your comments, suggestions and critics are below.

Line 55. “switching” and “more advantageous” are loaded words here. Why not acknowledge that certain lineages evolved to exploit new environments without making judgements about which lifestyles are best. There are still many flagellate fungi and unicellular fungi alive and well in aquatic and terrestrial environments.

We changed that sentence to: “The first ‘land’ fungi retained their flagella for motility and remained adapted to life in water, and several diverging lineages switched to multicellularity while exploiting new environments [1].”

Line 60. Delete “the” before “reproductive” – deleted.

Lines 61-62. Does it follow that reproductive modes in the zygomycetes and lack of macroscopic fruiting bodies make these organisms more difficult to study than members of the dikarya?

“making them more difficult to study than Dikarya.” Removed.

Figure 2 legend and panels. The panels are not labeled A and B, but these designations occur in the legend. I suggest adding the labels to the figures or changing the legend to refer to upper and lower panels.

Legend changed: Figure 2. Approaches used in the generation of zygomycete genomes; sequencing (upper panel) and assembly (lower panel) technologies.

Figure 3. Same comment about letters as for Figure 2.

Figure 3 legend. Change “sequences zygomycetes” to “sequences of zygomycetes” or “zygomycete sequences”. Also, delete “It covers” (its meaning is unclear and it is not necessary).

Legend changed: “Figure 3. Genome statistics for zygomycete sequences. Sizes of the genomes (upper left), number of contigs in assemblies (upper right), estimated sequencing depth coverage relative to the single genome (lower left), and the number of the predicted genes (lower right).”

Line 162. Delete “the” before “contigs”. – deleted.

Line 190. Change “the patient” to “a patient” – changed.

Line 202. Change “analysis” to “analyses” – changed.

Line 212. Perhaps change “turned to be” to “has involved” – changed.

Line 226. Change “that transposon” to “for transposon” – changed.

Line 310. Should “coexisting two sets” be changed to “two sets of coexisting”? – changed.

Line 311. Should the comma after colonization be replaced with a dash to make it complement the dash in line 310?

Replaced.

Line 401. It is not clear what the numbers in the following parenthetical expression refer to: “(e.g., 50 or 100)” – we removed this technical (e.g., 50 or 100) mentioning.

Line 416. A space is needed in “fromthe”. – space inserted.

Lines 457-464. This section seems to assume the reader will already know about the Phycomyces mad genes. I think it should be fleshed out a bit, and maybe the relationships these genes have to the wc genes of Neurospora and/or other members of the Dikarya should be clarified.

We completely rewrite second paragraph of this part of the manuscript and now it is better readable.

502-535 Sensing and responding to light have been explored in the model Mucorales species Phycomyces blakesleeanus. Mutants impaired in photoresponses, especially phototropism, were isolated as reported in the 1970s [142]. However, the first genes related to photoresponses were not reported until 2006 [143]. These were identified based on a candidate gene approach in which P. blakesleeanus was assumed to have similar blue light sensors as the Dikarya species [143,144], wherein a two-protein complex had been first identified in Neurospora crassa, defining a sensing and signaling system that uses a light-sensing domain coupled to transcription factor roles for signal transmission. With the release of the P. blakesleeanus genome sequence in 2010, identifying a candidate gene for madB as encoding the interacting protein was done using BLAST against the genome sequence [145]. The genome sequence was performed along with that of M. circinelloidies and revealed an expansion of this pair of genes required for light responses (3-4 copies of each). Being able to make targeted gene mutations in M. circinelloides of these homologs, as based of genome sequence information, provided a clear example of subfunctionalization after gene duplication [146]. The identification of the mutations in madI and madC mutants in P. blakesleeanus used traditional genetic mapping with molecular markers derived from the comparison of the genomes between two strains. With the regions narrowed, candidate genes could be identified and confirmed by sequencing [64,147]. These genes are conserved play roles in photobiology in the Dikarya species, suggesting that this innovation for terrestrial life had already evolved in early diverging fungal lineages. For instance, phenotypic analysis of the P. blakesleeanus madA and madB mutants indicates this protein pair has a conserved function in protecting fungi from harmful ultraviolet light [148].

Line 508. The phrase “not only on endohyphal level” is not clear. – “not only on endohyphal level” is removed. The meaning of the sentence is clearer without it.

Lines 524-526. This sentence is not clear: “The secretome composition of basal fungal lineages is shaped when nutrition models diversified early in their evolution.”

We removed this sentence from the text.

Line 574. Perhaps change “expected” to “expect.” – changed.
